# Height and Active Arterial Wall Thickening in Relation to Thyroid Cysts Status among Elderly Japanese: A Prospective Study

**DOI:** 10.3390/biology11121756

**Published:** 2022-12-02

**Authors:** Yuji Shimizu, Shin-Ya Kawashiri, Yuko Noguchi, Seiko Nakamichi, Yasuhiro Nagata, Takahiro Maeda, Naomi Hayashida

**Affiliations:** 1Department of General Medicine, Graduate School of Biomedical Sciences, Nagasaki University, Nagasaki 852-8501, Japan; 2Department of Cardiovascular Disease Prevention, Osaka Center for Cancer and Cardiovascular Diseases Prevention, Osaka 537-8511, Japan; 3Department of Community Medicine, Graduate School of Biomedical Sciences, Nagasaki University, Nagasaki 852-8523, Japan; 4Leading Medical Research Core Unit, Graduate School of Biomedical Sciences, Nagasaki University, Nagasaki 852-8523, Japan; 5Nagasaki University Health Center, Nagasaki 852-8521, Japan; 6Division of Strategic Collaborative Research, Center for Promotion of Collaborative Research on Radiation and Environment Health Effect, Atomic Bomb Disease Institute, Nagasaki University, Nagasaki 852-8523, Japan

**Keywords:** active arterial wall thickening, atherosclerosis, CIMT, endothelial repair, height, thyroid cyst

## Abstract

**Simple Summary:**

Since active endothelial repair leads to arterial wall thickening, as defined based on the degree of yearly progression in arterial wall thickness, arterial wall thickening could indicate endothelial repair activity. Recent studies indicate that individuals with short stature might have disadvantages in endothelial healing. Therefore, the need for endothelial repair should be much stronger in individuals with short versus tall stature. Thus, height could be inversely associated with arterial wall thickness. Aging is a process associated with declining capacity for endothelial repair. In addition, as thyroid hormone plays an important role in activating endothelial repair, only individuals with enough capacity to produce thyroid hormone can activate endothelial repair efficiently. These findings indicate that height could be inversely associated with active arterial wall thickening among participants without latent thyroid damage. Our previous studies with thyroid ultrasound revealed that the absence of thyroid cysts could indicate the presence of latent damage of the ability to produce thyroid hormone. In this study, a significant inverse association between height and active arterial wall thickening was observed only among participants with thyroid cysts; this finding can help clarify a novel mechanism in vascular remodeling.

**Abstract:**

Height is inversely associated with inflammation that stimulates endothelial repair. In our previous study involving elderly men aged 60–69 years, we found that active arterial wall thickening, which is known as the process of endothelial repair, requires CD34-positive cells. As thyroid hormone regulates CD34-positive cell production and as the absence of thyroid cysts might indicate latent damage in the thyroid, the status of thyroid cysts possibly influences the association between height and active arterial wall thickening. We conducted a 2-year follow-up study of Japanese aged 60–69 years. For participants with thyroid cysts, height was significantly inversely associated with active arterial wall thickening (thyroid function and baseline CIMT adjusted odds ratio of active arterial wall thickening for one increment of standard deviation of height (5.7 cm for men and 4.8 cm for women), 0.66 [0.49, 0.89]), while for those without thyroid cysts, a positive tendency between the two parameters was observed (1.19 [0.96, 1.50]). An inverse association between height and active arterial wall thickening was observed only for elderly participants with thyroid cysts possibly because of a supportive role of thyroid hormone, as the absence of thyroid cysts might indicate latent damage in the thyroid.

## 1. Introduction

Recently, both inflammation [1] and hypertension [2] have been reported to be inversely associated with height. Inflammation and hypertension activate endothelial repair by inducing endothelial injury. There are strong associations between active endothelial repair and active arterial wall thickening; active arterial wall thickening indicates the presence of active endothelial repair. Thus, height could be inversely associated with active arterial wall thickening by indicating the presence of active arterial repair. 

Hematopoietic stem cells such as CD34-positive cells directly contribute to endothelial repair [3,4,5]. A prospective study of elderly men clarified that circulating CD34-positive cell are required for active arterial wall thickening [6]. However, other previous studies on elderly men have indicated that the productivity of these cells might be positively associated with height [7,8]. Therefore, only among elderly participants with sufficient capacity to produce CD34-positive cells, height could be inversely associated with active arterial wall thickening.

In addition, thyroid hormone regulates the proliferation of CD34-positive cells [9]. The existence of thyroid cysts could indicate the absence of latent thyroid damage [10,11,12]. Anti-thyroid peroxidase antibodies are known to decrease thyroid hormone synthesis. A cross-sectional study of Japanese with normal thyroid function reported an inverse association between anti-thyroid peroxidase antibody titer and thyroid cysts [10]. Anti-thyroid peroxidase antibody titers were also reported to be inversely associated with active arterial wall thickening [13]. Therefore, participants without thyroid cysts might have worse endothelial health. In other words, among older individuals, the presence of thyroid cysts could indicate a relatively higher capacity for producing CD34-positive cells. 

Since height has been reported to be inversely associated with death from all causes and cardiovascular disease [14] and circulating CD34-positive cells have been reported to be inversely associated with cardiovascular disease and all-cause mortality [15,16], understanding the influence of thyroid cysts on the association between height and active arterial wall thickening could help identify a novel mechanism in vascular remodeling. 

We hypothesized that the inverse association between height and active arterial wall thickening might be observed only in elderly participants with thyroid cysts. To study this hypothesis, we conducted a 2.9-year prospective study of elderly people aged 60–69 years.

## 2. Materials and Methods

### 2.1. Study Population

The present study was performed in the town of Saza, Nagasaki prefecture [10,11,12,13,17]. The town of Saza is located in Kyusyu district in Japan. In 2015, 2113 residents of Saza were aged 60–69 years [18]. Written consent forms were made available to ensure that the participants understood the study objectives. Informed consent was obtained from all participants.

This study was approved by the Ethics Committee of Nagasaki, University Graduate School of Biomedical Sciences (Project registration number: 14051404-13).

The study population comprised 817 Japanese individuals aged 60–69 years from Saza town in the western part of Japan who underwent an annual health check-up in 2014.

Participants with a history of thyroid disease (*n* = 29) were excluded from the study population for the purpose of removing the influence of thyroid disease. Participants without height (*n* = 1) or thyroid function data such as thyroid-stimulating hormone (TSH), free triiodothyronine (T3), and free thyroxine (T4) data (*n* = 11) were also excluded. Additionally, participants who did not undergo our annual health check-up between 2015 and 2017 were excluded (*n* = 36). A total of 740 participants were included in the study. The mean age of those participants was 65.2 years with standard deviation [SD] of 2.6.

### 2.2. Data Collection and Laboratory Measurements

Specially trained medical staff obtained information on the history of thyroid disease. By using standard procedures at the LSI Medience Corporation (Tokyo, Japan), thyroid function-related hormones (TSH, free T3, and free T4) were measured via chemiluminescence immunoassay.

Carotid intima-media thickness (CIMT) of the common carotid arteries was measured by experienced vascular examiners using ultrasound equipment (LOGIQ Book XP with a 10-MHz transducer; GE Healthcare, Milwaukee, WI, USA). Maximum values for left and right common CIMT were calculated using digital edge-detection software (IntimaScope; MediaCross, Tokyo, Japan), via a previously described protocol [19]. The recently developed IntimaScope software was used to increase the accuracy and reproducibility of CIMT measurement values. This software semi-automatically recognizes the edges of the internal and external membranes of the artery and automatically determines the distance at a sub-pixel level (estimated to be 0.01 mm) [20]. 

Baseline subclinical atherosclerosis was diagnosed as a CIMT of ≥1.1 mm because a normal CIMT value has been previously reported to be <1.1 mm [21]. Given that the present study used IntimaScope, we defined active arterial wall thickening as increased values of CIMT ≥ 0.01 mm/year, as in our previous studies [6,13].

Additionally, thyroid cysts were detected by experienced technicians using a LOGIQ Book XP with a 10-MHz transducer (GE Healthcare, Milwaukee, WI, USA). In this study, a cyst (maximum diameter ≥ 2.0 mm) in the thyroid without a solid component was defined as a thyroid cyst, in accordance with our previous studies [10,11,12].

### 2.3. Statistical Analysis

Except for TSH and CIMT, clinical characteristics of the study population based on height tertile were expressed as mean ± SD. Because TSH and CIMT showed a skewed distribution, they were expressed as median (first quartile, third quartile), followed by logarithmic transformation. Significant differences between height levels were evaluated by using analysis of variance. 

Height tertiles for men and women are as follows: <162.5 and <150.6 cm for Tertile 1 (low), 162.5–167.1 and 150.6–154.8 cm for Tertile 2 (middle), and ≥167.2 and ≥154.9 cm for Tertile 3 (high), respectively.

To evaluate the correlation between height and age, sex-adjusted partial correlation coefficients (r) were calculated. Logistic regression models were used to calculate odds ratios and 95% confidence intervals and to identify associations between active arterial wall thickening and baseline atherosclerosis and between height and active arterial wall thickening. The goodness of fit of all logistic regression models was evaluated using the Hosmer–Lemeshow test.

The present study aimed to clarify the influence of thyroid cysts on the association between active arterial wall thickening and height. We hypothesized that thyroid cysts might act as an indicator of decreased demand for thyroid hormone that results in relatively higher thyroid hormone activity. This relative increase in thyroid hormone activity might have a beneficial effect on endothelial repair. Therefore, baseline thyroid function and baseline CIMT could act as confounders in the present study. 

Four models were used for analysis. Model 1 was an unadjusted model. Model 2 was adjusted for only sex and age; model 3 was adjusted for sex, age, free T3 (pmol/L), and TSH (logarithmic value); and model 4 was additionally adjusted for baseline CIMT (logarithmic value). Models 1, 2, and 3 were used for the analyses of the associations between active arterial wall thickening and baseline atherosclerosis, and height and active arterial wall thickening, while model 4 was used to analyze only the association between height and active arterial wall thickening.

The SAS system for Windows (version 9.4; SAS Inc., Cary, NC, USA) was used for all statistical analyses performed in this study. Statistical significance was considered at a *p* value of less than 0.05.

## 3. Results

Among the 740 Japanese individuals aged 60–69 years, 96 (13.0%) and 306 (41.4%) individuals showed atherosclerosis at baseline (baseline atherosclerosis) and active arterial wall thickening during follow-up, respectively.

No significant correlations between height and age were observed overall, among participants without thyroid cysts, or among participants with thyroid cysts. In the partial correlation analysis, the sex-adjusted values were r = −0.04 (*p* = 0.277) overall, r = −0.04 (*p* = 0.346) among participants without thyroid cysts, and r = −0.04 (*p* = 0.557) among participants with thyroid cysts.

### 3.1. Clinical Characteristics of the Study Population

The clinical characteristics of the study population by tertile of height level are shown in Table 1. Height was revealed to be significantly inversely associated with age and TSH only among participants overall.

### 3.2. Association between Active Arterial Wall Thickening and Baseline Atherosclerosis

Significant inverse associations between active arterial wall thickening and baseline atherosclerosis were shown in the crude model (model 1) and the sex and age-adjusted model (model 2). This inverse association remained significant even after further adjustment for free T3 and TSH (model 3) (Table 2).

### 3.3. Association between Active Arterial Wall Thickening and Height

For all participants, no significant association between height and active arterial wall thickening was observed. Further analyses stratified by thyroid cysts status showed that participants without thyroid cysts exhibited a positive tendency of association between height and active arterial wall thickening, while participants with thyroid cysts exhibited a significant inverse association between the parameters (Table 3).

Further, the influence of thyroid cysts status on the associations between height levels and active arterial wall thickening were shown to be significant. The adjusted *p*-values for this interaction were 0.003, 0.002, 0.005, and 0.003 for models 1, 2, 3, and 4.

### 3.4. Additional Sensitivity Analysis

For additional sensitivity analyses, sex-specific associations between active arterial wall thickening and height were evaluated; the associations observed were essentially the same. In Model 3, the odds ratio and 95% confidence interval of active arterial wall thickening for a 1 standard deviation increment in height among participants without and with thyroid cysts were 1.30 (0.92, 1.82) and 0.76 (0.44, 1.32) in men and 1.12 (0.85, 1.47) and 0.62 (0.44, 0.89) in women, respectively.

## 4. Discussion

In this study, significant inverse association between height and active arterial wall thickening is observed in elderly participants with thyroid cysts but not in those without thyroid cysts.

Many studies have demonstrated an inverse association between height and cardiovascular disease [14,22,23] and an inverse association between height and hypertension [2]. Therefore, individuals with short stature might have a higher risk of endothelial injury, which induces stronger endothelial repair activity, than individuals with tall stature. A previous study involving older men aged 60 to 69 years showed an inverse association between height and active endothelial repair only for those with enough hematopoietic activity as evaluated by hemoglobin level [24]. Our present results were partly compatible with those findings; we observed an inverse association between height and active arterial wall thickening among participants with thyroid cysts. 

Height is known to be strongly inversely associated with age because aging is a process that decreases height [2,25,26]. Aging could influence the risk of endothelial dysfunction [27]. Participants with thyroid cysts are older than those without thyroid cysts [11,12,28]. Therefore, age could have a strong influence on the association between height and active arterial wall thickening. To avoid the influence of age in the present study, we limited participants to those aged 60 to 69 years. We did not find a significant correlation between height and age in this study population. Partial correlation analysis that adjusted for sex found no significant correlations between height and age overall or when participants were stratified by thyroid cyst status. Thus, age should have had a limited influence on the association among height, active arterial wall thickening, and thyroid cysts in the present study.

Figure 1 shows the potential mechanism underlying our results. Identified associations in the present study are shown in red [Figure 1a–c] As a result of the presence of low hematopoietic activity [17,24,29,30], dyslipidemia [31], and chronic low-grade inflammation-related genetic factor [32], for elderly participants with short stature, stronger endothelial injury than for those with tall stature could be observed.

Endothelial injury induces endothelial repair, which relates to active arterial wall thickening. Further, active arterial wall thickening requires CD34-positive cells [6]. As the presence of baseline atherosclerosis could cause a shortage of circulating CD34-positive cells due to heavy consumption, as in our previous study, a significant inverse association between baseline atherosclerosis and active arterial wall thickening was observed (Table 2, Figure 1a) [6]. A cohort study with 36,984 participants followed for 7.0 years reported no association between yearly CIMT progression and cardiovascular disease [33]. However, atherosclerosis evaluated by CIMT [34] is a significant cardiovascular risk factor with global consensus [35]. Those studies support our present finding that shows a significant inverse association between baseline atherosclerosis and active arterial wall thickening.

Aging is a process that decreases physical activity and reduces the demand for thyroid hormone because thyroid hormone regulates energy metabolism. Additionally, participants with thyroid cysts are found to be older than those without thyroid cysts [11,12,28]. Then, during the process of aging, with the process of reduced demand for thyroid hormone, thyroid cysts could be formed. 

Therefore, even though serum concentrations of thyroid hormone are essentially the same among participants with and without thyroid cysts, participants with thyroid cysts might have relatively higher thyroid hormone activity than those without thyroid cysts because the demand for thyroid hormone is lower in patients with thyroid cysts. A previous study reported that HbA1c levels were higher in non-diabetic participants with subclinical hypothyroidism than in normal healthy controls [36]. One of our previous studies of euthyroid individuals revealed that an inverse association between HbA1c and thyroid cysts partly explains this concept; participants with thyroid cysts might have relatively higher thyroid hormone activity than those without thyroid cysts [37]. 

Furthermore, the absence of thyroid cysts might act as an indicator of latent damage of the thyroid, which decreases the productivity of thyroid hormone [10,11,12]. Participants who do not have thyroid cysts might have comparatively lower capacity to produce thyroid hormone. Therefore, among participants without thyroid cysts, even when thyroid hormone production by the entire thyroid gland remains similar to that of participants with thyroid cysts, production per unit of remaining healthy thyroid gland might be higher because of latent thyroid damage. Higher demand for thyroid hormone per unit of remaining healthy thyroid gland might reduce the number and size of thyroid cysts. Further development of methods to evaluate thyroid demand is necessary to clarify the influence of thyroid hormone demand on thyroid cyst formation. Renal function could be a candidate for evaluating the level of demand for thyroid hormone [38]. 

TSH might indicate thyroid hormone activity among participants with enough capacity to produce thyroid hormone, whereas TSH might indicate deficient thyroid hormone activity among those without enough capacity to produce thyroid hormone. A previous Japanese study reported that the prevalence of overt hypothyroidism and subclinical hypothyroidism was higher among patients with a higher chronic kidney disease (CKD) stage with increased levels of urinary protein [39]. Proteinuria is a major risk factor for CKD [40] and high-normal levels of TSH is a significant risk factor for the development of CKD [41]. Therefore, our previous study that reported a significant inverse association between TSH and proteinuria in euthyroid participants with thyroid cysts but a significant positive association between TSH and proteinuria in euthyroid participants without thyroid cysts [42] could explain this concept. Details on the concept that euthyroid participants with thyroid cysts might have comparative higher thyroid hormone activity than euthyroid patients without thyroid cysts have been described elsewhere [28,42].

Notably, thyroid hormone reportedly regulates the proliferation of CD34-positive cells [9]. Even among euthyroid participants, those with thyroid cysts might have relatively higher thyroid hormone activity than those without thyroid cysts [28,42]. Then, participants with thyroid cysts might experience a beneficial influence on active arterial wall thickening. Among middle-aged Japanese men (40–59 years) with body mass index (BMI) ≥ 23 kg/m^2^, height was reported to be significantly inversely associated with low-grade inflammation [1]. BMI has been shown to be positively associated with low-grade inflammation among middle-aged Japanese (40–69 years) [43]. Low-grade inflammation induces vascular aging [44], which also increases the necessity of vascular repair. Since active arterial wall thickening is observed as a process of vascular repair, inverse association between height and active arterial wall thickening could be observed in the elderly. 

In the present study, a significant inverse association between active arterial wall thickening and height was observed in elderly participants with thyroid cysts (Table 3, Figure 1b).

Thyroid hormone regulates the proliferation of CD34-positive cells [9]. For active arterial wall thickening, a sufficient number of circulating CD34-positive cells is mandatory [6]. Then, elderly patients with thyroid cysts might have enough CD34-positive cells to activate endothelial repair, whereas those without thyroid cysts might not.

Among general Japanese individuals with normal levels of free T3 and free T4, anti-thyroid peroxidase antibody titers were inversely associated with thyroid cysts [10] and active arterial wall thickening [13], which requires CD34-positive cells [6]. Therefore, low-grade inflammation, which is related to anti-thyroid peroxidase antibody titers, might reduce both the presence of thyroid cysts and endothelial repair activity.

However, the reason why height is inversely associated with inflammation [1] is unclear. Anemia has been reported to be associated with growth delay in children [45], while height is inversely associated with anemia in adults [30]. Additionally, reticulocytes are inversely associated with height among elderly individuals [24,29]. Since reticulocytes have been shown to be inversely associated with atherosclerosis [46], low activity of anti-oxidative effect that relates to low production of these cells might be responsible for the inflammatory disadvantage that short-stature individuals might possess. In addition, inflammation-related genetic factors [32,47] and dyslipidemia [31] are also associated with short stature. These factors might be responsible for the inverse association between height and low-grade inflammation. Furthermore, previous studies showed a significant association between genetically determined shorter height and a higher risk of cardiovascular disease [48,49]. However, no studies have reported an association between those genetic factors and thyroid cysts.

A significant positive association between circulating CD34-positive cells and height among elderly men aged 65–69 years was also reported in our previous study [7,8]. Aging is a process that deteriorates the activity of hematopoietic stem cells [50], with hematopoietic bone marrow activity declining with age [51,52,53,54]. Since height could act as an indicator of the absolute volume of bone marrow, compared to elderly participants with tall stature, the negative influence of aging on hematopoietic activity could be critical for those with short stature. Then, height could act as an indicator of the capacity of endothelial repair in the elderly. The absence of thyroid cysts that are associated with latent thyroid damage [10,11,12] might be associated with low proliferation of CD34-positive cells [9]. Further, height might determine CD34-positive cell production activity among elderly participants without thyroid cysts. Therefore, even though statistical significance was not shown, a positive tendency between height and active arterial wall thickening was observed among elderly participants without thyroid cysts in the present study (Table 3, Figure 1c).

Since height is inversely associated with cardiovascular disease [23,55,56], while being positively associated with cancer [57,58,59], evaluating thyroid cyst and height could be an efficient tool for the risk estimation of cardiovascular disease and cancer. In general, thyroid cysts have been regarded as not clinically significant. However, because the influence of thyroid cysts on the association between height and arterial wall thickening was shown to be significant, the present study revealed that the presence of thyroid cysts could play an important role in endothelial repair. Further investigation with data on these diseases is necessary.

Potential limitations of this study warrant consideration. Even though circulating CD34-positive cells might have the main role in the present associations, we could not obtain data on these cells because of measurement difficulty during annual health check-up. In addition, because of the limited sample size of the present study, no significant association was observed when the association was analyzed in the subset of participants without thyroid cysts. Further investigation with a larger sample size of study population is necessary. In additional sensitivity analyses, we found essentially the same associations as in the main analysis. In men with thyroid cysts, no significant association between height and active arterial wall thickening was observed. Therefore, to perform statistically meaningful sex-specific analysis, further investigations that involve more men are necessary.

## 5. Conclusions

In elderly individuals with thyroid cysts, height was found to be significantly inversely associated with active arterial wall thickening, while for those without thyroid cysts, although no statistical significance was exhibited, it showed a positive tendency of association between the parameters. Therefore, thyroid cysts might have a beneficial effect on endothelial repair activity while short stature is an independent risk factor for endothelial injury, which induces active arterial wall thickening. These findings indicate a novel mechanism underlying vascular remodeling related to latent damage of the thyroid and height among elderly Japanese individuals.

## Figures and Tables

**Figure 1 biology-11-01756-f001:**
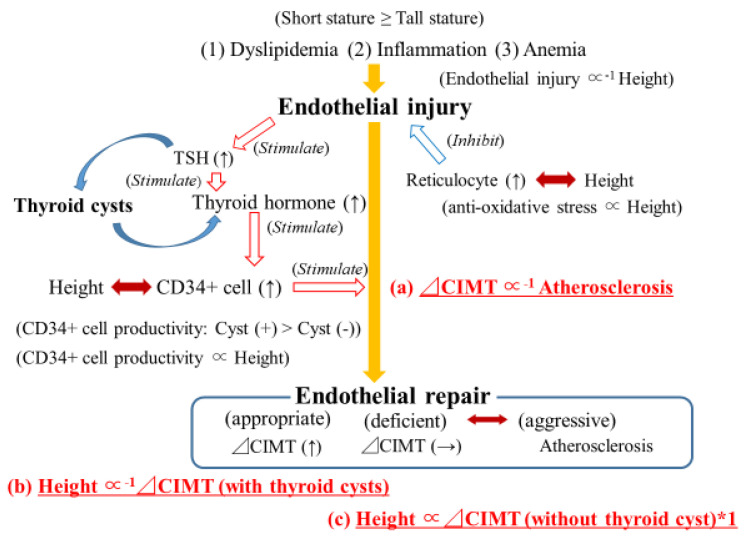
Potential mechanisms underlying the association between height, baseline atherosclerosis, and active arterial wall thickening. Atherosclerosis: baseline atherosclerosis. CIMT: carotid intima-media thickness. ⊿CIMT: active arterial wall thickening. Associations coded in red were identified in the present study (**a**–**c**). *1: The association was not statistically significant.

**Table 1 biology-11-01756-t001:** Clinical characteristics of study participants by height level.

		Height Tertile			*p*
		Tertile 1 (Low)	Tertile 2 (Middle)	Tertile 3 (High)	
Total					
	No. of participants	244	251	245	
	Men, %	40.6	39.8	40.8	0.974
	Age, year	65.4 ± 2.7	65.3 ± 2.6	64.8 ± 2.5	0.047
	free T3, pmol/L	4.87 ± 0.48	4.90 ± 0.52	4.86 ± 0.51	0.663
	free T4, pmol/L	0.16 ± 0.02	0.16 ± 0.02	0.16 ± 0.02	0.429
	TSH, mIU/L	1.72 [1.23, 2.48] *^1^	1.53 [1.10, 2.35] *^1^	1.47 [1.08, 2.31] *^1^	0.044 *^2^
	CIMT, mm	0.86 [0.75, 0.97] *^1^	0.84 [0.76, 0.96] *^1^	0.85 [0.77, 0.99] *^1^	0.370 *^2^
Thyroid cyst (−)					
	No. of participants	164	157	151	
	Men, %	49.3	49.9	50.0	0.466
	Age, year	65.4 ± 2.7	65.2 ± 2.5	64.8 ± 2.6	0.211
	free T3, pmol/L	4.88 ± 0.45	4.86 ± 0.55	4.93 ± 0.50	0.392
	free T4, pmol/L	0.16 ± 0.03	0.16 ± 0.03	0.16 ± 0.02	0.707
	TSH, mIU/L	1.70 [1.22, 2.41] *^1^	1.47 [1.09, 2.37] *^1^	1.48 [1.09, 2.35] *^2^	0.260 *^2^
	CIMT, mm	0.85 [0.75, 0.97] *^1^	0.84 [0.76, 0.95] *^1^	0.85 [0.76, 0.98] *^2^	0.809 *^2^
Thyroid cyst (+)					
	No. of participants	80	94	94	
	Men, %	49.3	46.4	46.0	0.304
	Age, year	65.3 ± 2.7	65.5 ± 2.7	64.8 ± 2.4	0.140
	free T3, pmol/L	4.86 ± 0.53	4.98 ± 0.47	4.75 ± 0.50	0.008
	freeT4, pmol/L	0.16 ± 0.02	0.15 ± 0.02	0.16 ± 0.02	0.331
	TSH, mIU/L	1.75 [1.36, 2.87] *^1^	1.67 [1.12, 2.35] *^2^	1.45 [1.08, 2.22] *^1^	0.118 *^2^
	CIMT, mm	0.88 [0.75, 0.98] *^1^	0.86 [0.77, 0.99] *^1^	0.85 [0.78, 1.04] *^1^	0.370 *^2^

T3: triiodothyronine, T4: thyroxine, TSH: thyroid stimulating hormone, CIMT: carotid intima-media thickness. Values are mean ± standard deviation. Height tertiles among men were <162.5 cm for Tertile 1 (low), 162.5–167.1 cm for Tertile 2 (middle), and ≥167.2 cm for Tertile 3 (high). Among women, the corresponding values were <150.6 cm, 150.6–154.8 cm, and ≥154.9 cm. *^1^: Values are median [the first quartile, the third quartile]. *^2^: Logarithmic transformation was used for evaluating *p*.

**Table 2 biology-11-01756-t002:** Association between active arterial wall thickening and baseline atherosclerosis.

	Baseline Atherosclerosis		*p*
	(−)	(+)	
No. of participants	644	96	
No. of cases (%)	299 (46.4)	7 (7.3)	
Model 1	Ref	0.09 (0.04, 0.20)	<0.001
Model 2	Ref	0.09 (0.04, 0.20)	<0.001
Model 3	Ref	0.09 (0.04, 0.20)	<0.001

Ref: Reference. Model 1: unadjusted. Model 2: adjusted only for sex and age. Model 3: further adjusted for free triiodothyronine and thyroid-stimulating hormone.

**Table 3 biology-11-01756-t003:** Associations between height and active arterial wall thickening among participants overall and stratified by thyroid cyst status.

	Height Tertile			*p* for Trend	1SD Increment in Height (5.7 cm for Men and 4.8 cm for Women)
	Tertile 1 (Low)	Tertile 2 (Middle)	Tertile 3 (High)		
Total					
No. of participants	244	251	245		
No. of cases (%)	105 (43.0)	100 (39.8)	101 (41.2)		
Model 1	Ref	0.88 (0.61, 1.25)	0.93 (0.65, 1.33)	0.685	0.94 (0.81, 1.09)
Model 2	Ref	0.88 (0.61, 1.25)	0.92 (0.64, 1.32)	0.663	0.94 (0.81, 1.09)
Model 3	Ref	0.87 (0.61, 1.24)	0.91 (0.64, 1.31)	0.620	0.94 (0.81, 1.09)
Model 4	Ref	0.86 (0.58, 1.27)	0.99 (0.67, 1.46)	0.941	0.99 (0.83, 1.16)
Thyroid cyst (−)					
No. of participants	164	157	151		
No. of cases (%)	62 (37.8)	63 (40.1)	70 (46.4)		
Model 1	Ref	1.11 (0.70, 1.73)	1.42 (0.91, 2.23)	0.126	1.12 (0.93, 1.35)
Model 2	Ref	1.10 (0.71, 1.73)	1.43 (0.91, 2.24)	0.126	1.12 (0.93, 1.35)
Model 3	Ref	1.11 (0.71, 1.74)	1.43 (0.91, 1.74)	0.126	1.12 (0.93, 1.35)
Model 4	Ref	1.09 (0.67, 1.77)	1.62 (0.98, 2.68)	0.061	1.19 (0.96, 1.50)
Thyroid cyst (+)					
No. of participants	80	94	94		
No. of cases (%)	43 (53.8)	37 (39.4)	31 (33.0)		
Model 1	Ref	0.56 (0.31, 1.02)	0.42 (0.23, 0.78)	0.007	0.69 (0.57, 0.89)
Model 2	Ref	0.54 (0.29, 0.995)	0.40 (0.21, 0.75)	0.004	0.67 (0.51, 0.86)
Model 3	Ref	0.48 (0.26, 0.91)	0.32 (0.17, 0.62)	<0.001	0.62 (0.48, 0.82)
Model 4	Ref	0.47 (0.24, 0.94)	0.33 (0.16, 0.67)	0.002	0.66 (0.49, 0.89)

Ref: Reference. Model 1: unadjusted. Model 2: adjusted for only sex and age. Model 3: further adjusted for free triiodothyronine and thyroid-stimulating hormone. Model 4: additionally adjusted for baseline carotid intima-media thickness. Height tertiles among men were <162.5 cm for Tertile 1 (low), 162.5–167.1 cm for Tertile 2 (middle), and ≥167.2 cm for Tertile 3 (high). Among women, the corresponding values were <150.6 cm, 150.6–154.8 cm, and ≥154.9 cm.

## Data Availability

We cannot publicly provide individual data due to participant privacy, according to ethical guidelines in Japan. Additionally, the informed consent obtained does not include a provision for publicity-sharing data. Qualifying researchers may apply to access a minimal dataset by contacting Prof Naomi Hayashida, Principal Investigator, Division of Promotion of Collaborative Research on Radiation and Environment Health Effects, Atomic Bomb Disease Institute, Nagasaki University, Nagasaki, Japan at naomin@nagasaki-u.ac.jp. Or, please contact the office of data management at ritouken@vc.fctv-net.jp. Information for where data request is also available online: https://www.genken.nagasaki-u.ac.jp/dscr/message/ and http://www.med.nagasaki-u.ac.jp/cm/ (accessed on 9 February 2022).

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
