# Peer review of "Height and Active Arterial Wall Thickening in Relation to Thyroid Cysts Status among Elderly Japanese: A Prospective Study"

_biology, 2022, doi:10.3390/biology11121756_

Round 1
Reviewer 1 Report
Dear Authors,
A very interesting study, well constructed with a clinical impact. Some major points for revision. Kindly describe the sample randomization. Why exists the age limitation? It would be of great interest if your study could extend to other age groups (as you refer even younger population 40 and over years of age). The gender impact also would be of interest to be examined.
Delete all details from your previous work from the abstract section. It is better to construct a such paragraph into the discussion section.
the discussion should be better constructed (studies with published data on this field, as well as clearly your contribution).
A more streight conclusion would be of great impact for such a study.
Study limitations is a session dynamic and not potential. Kindly refer clearly the main limitation of the sample size, as well as other potential subgroup analysis.
Also, a comment of other thyroid pathology and its impact would be of great interest, as the variable of inflammation contributes to the major part of pathological entities.
Thank you
Author Response
Reviewer 1
1)Why exists the age limitation? It would be of great interest if your study could extend to other age groups (as you refer even younger population 40 and over years of age). The gender impact also would be of interest to be examined.
→
(About the range of age)
Thank you for valuable comment. According to this reviewer’s valuable comment, I re-checked the reason why we limited the target population for narrow range of age.
When considering the association between height and active arterial wall thickening in relation to status of thyroid cysts, influence of age should be taken into consideration because of the following reasons.
Height is well known factor that is strongly inversely associated with age because aging is the process that decrease the height level [Ref1][Ref2]. Then age could influence on height levels. And because aging is also known process that increase the risk of endothelial dysfunction [Ref3], age could influence on active arterial wall thickening. In addition to that participants with thyroid cysts are found to be older than those without thyroid cysts [12,13,27]. Therefore, age could strong influence on the association between height and active arterial wall thickening. Age could influence on those targeted variants. Therefore, even we used age-adjusted model the model became under-adjusted when we analyze among participants with wide range of age. This is the reason why present study requires a well-thought-out research plan.
And when we evaluate the correlation between height and age among participants aged 40 to 69 years, age revealed to be significantly inversely correlated with height for both those without thyroid cysts (sex adjusted partial correlation coefficient (r)=-0.35, p<0.001) and with thyroid cysts (r=-0.32, p<0.001). And when we evaluate those correlation among present study population, no significant correlation was observed. The adjusted correlation coefficients (p) were r=-0.04 (p=0.346) for those without thyroid cysts and r=-0.04 (p=0.557) for those with thyroid cyst. Then present study population is appropriate for present study.
[Ref1]
Shimizu, Y.; Kawashiri, S.Y.; Nobusue, K.; Nonaka, F.; Tamai, M.; Honda, Y.; Yamanashi, H.; Nakamichi, S.; Kiyama, M.; Hayashida, N.; et al.; Association between circulating CD34-posistive cell count and height loss among older men. Sci Rep. 2022, 12(1), 7175. doi: 10.1038/s41598-022-11040-y.
[Ref2]
Shimizu, Y.; Hayakawa, H.; Takada, M.; Okada, T.; Kiyama, M.; Hemoglobin and adult height loss among Japanese workers: A retrospective study. PLoS. One 2021, 16(8), e0256281. doi: 10.1371/journal.pone.0256281.
[Ref3]
Al-Shaer, M.H.; Choueiri, N.E.; Correia, M.L.; Sinkey, C.A.; Barenz, T.A.; Haynes, W.G.; Effect of aging and atherosclerosis on endothelial and vascular smooth muscle function in humans. Int. J. Cardiol 2006, 109(2), 201-6. doi: 10.1016/j.ijcard.2005.06.002. Epub 2005 Jul 27.
Since present study population is 60 to 69 years, we revised as following in method section.
In 2015, 2,187 residents of Saza were aged 60–69 years [18].
Then we added following sentences in manuscript
In statistical analysis section
To evaluate the correlation between height and age, sex-adjusted partial correlation coefficients (r) were calculated.
In result section
No significant correlations between height and age were observed overall, among participants without thyroid cysts, or among participants with thyroid cysts. In the partial correlation analysis, the sex-adjusted values were r=-0.04 (p=0.277) overall, r=-0.04 (p=0.346) among participants without thyroid cysts, and r=-0.04 (p=0.557) among participants with thyroid cysts.
In discussion section
Height is known to be strongly inversely associated with age because aging is a process that decreases height [2,25,26]. Aging could influence height levels by increasing the risk of endothelial dysfunction [27]. Participants with thyroid cysts are older than those without thyroid cysts [11,12,28]. Therefore, age could have a strong influence on the association between height and active arterial wall thickening. To avoid the influence of age in the present study, we limited participants to those aged 60 to 69 years. We did not find a significant correlation between height and age in this study population. Partial correlation analysis that adjusted for sex found no significant correlations between height and age overall or when participants were stratified by thyroid cyst status. Thus, age should have had a limited influence on the association among height, active arterial wall thickening, and thyroid cysts in the present study.
(About the gender)
Thank you for valuable comment. According to this reviewer’s valuable comment, I rechecked the additional sensitivity analysis as shown in result section.
Additional sensitivity analysis
For additional sensitivity analyses, sex-specific associations between active arterial wall thickening and height were evaluated; the associations observed were essentially the same. In Model 3, the odds ratio and 95% confidence interval of active arterial wall thickening for a 1 standard deviation increment in height among participants without and with thyroid cysts were 1.30 (0.92, 1.82) and 0.76 (0.44, 1.32) in men and 1.12 (0.85, 1.47) and 0.62 (0.44, 0.89) in women, respectively.
Since we found essentially same association between men and women, we concluded that there are no gender different associations on present results. And we also added following sentences in limitation section.
In additional sensitivity analyses, we found essentially the same associations as in the main analysis. In men with thyroid cysts, no significant association between height and active arterial wall thickening was observed. Therefore, to perform statistically meaningful sex-specific analysis, further investigations that involve more men are necessary.
2) Delete all details from your previous work from the abstract section. It is better to construct a such paragraph into the discussion section.
→
Thank you for valuable comment. According to this reviewer’s valuable comment, we delete the detail description about our previous study in abstraction section.
3) The discussion should be better constructed (studies with published data on this field, as well as clearly your contribution).
→
Thank you for valuable comment. According to this reviewer’s valuable comment, we reconsider the content of present discussion section. There are many studies that shows inverse association between height and cardiovascular disease [Ref4][Ref5] and inverse association between height and hypertension [2]. Therefore, individuals with short stature might have higher risk of endothelial injury that induce activity of endothelial repair than that with tall stature. Platelet count could indicate the activity of endothelial repair [Ref6] and inverse association between platelet count and height was observed among older men aged 60 to 69 years with enough hematopoietic activity evaluated by hemoglobin level [Ref7]. These studies indicates that height could be inversely associated with active endothelial repair among those with enough hematopoietic activity evaluated by hemoglobin level [Ref7][Ref8].
Therefore, individuals with short stature might have higher activity of endothelial repair than that with tall stature. Those studies are compatible with our present results that shows inverse association between height and active arterial wall thickening. Then we added following sentences in discussion section.
Many studies have demonstrated an inverse association between height and cardiovascular disease [14,22,23] and an inverse association between height and hypertension [2]. Therefore, individuals with short stature might have a higher risk of endothelial injury, which induces stronger endothelial repair activity, than individuals with tall stature. A previous study involving older men aged 60 to 69 years showed an inverse association between height and active endothelial repair only for those with enough hematopoietic activity as evaluated by hemoglobin level [24]. Our present results were partly compatible with those findings; we observed an inverse association between height and active arterial wall thickening among participants with thyroid cysts.
[Ref4]
Lee, C.M.; Barzi, F.; Woodward, M.; Batty, G.D.; Giles, G.G.; Wong, J.W.; Jamrozik, K.; Lam, T.H.; Ueshima, H.; Kim, H.C.; Gu, D.F.; et al.; Asia Pacific Cohort Studies Collaboration. Adult height and the risk of cardiovascular disease and major causes of death in the Asia-Pacific region: 21,000 deaths in 510,000 men and women. Int. J. Epidemiol 2009, 38(4), 1060-71. doi: 10.1093/ije/dyp150.
[Ref5]
Honjo, K.; Iso, H.; Inoue, M.; Tsugane, S.; Adult height and the risk of cardiovascular disease among middle aged men and women in Japan. Eur. J. Epidemiol 2011, 26(1), 13-21. doi: 10.1007/s10654-010-9515-8.
[Ref6]
Sato, S.; Koyamatsu, J.; Yamanashi, H.; Nagayoshi, M.; Kadota, K.; Maeda, T.; Platelets as an indicator of vascular repair in elderly Japanese men. Oncotarget 2016, 7(29):44919-44926. doi: 10.18632/oncotarget.10229.
[Ref7]
Shimizu, Y.; Sato, S.; Koyamatsu, J.; Yamanashi, H.; Nagayoshi, M.; Kadota, K.; Kawashiri, S.Y.; Maeda, T.; Possible mechanism underlying the association between height and vascular remodeling in elderly Japanese men. Oncotarget 2017, 23;9(8):7749-7757. doi: 10.18632/oncotarget.23660.
4) A more streight conclusion would be of great impact for such a study.
→
Thank you for valuable comment. According to this reviewer’s valuable comment, we reconsider about the conclusion of present study. And we added following sentences in conclusion section.
In elderly individuals with thyroid cysts, height found to be significantly inversely associated with active arterial wall thickening, while those without thyroid cysts, although exhibited no statistical significance, showed a positive tendency of association between the parameters. Therefore, thyroid cysts might have a beneficial effect on endothelial repair activity while short stature is an independent risk factor for endothelial injury, which induces active arterial wall thickening. These findings indicate a novel mechanism underlying vascular remodeling related to latent damage of the thyroid and height among elderly Japanese individuals.
5) Study limitations is a sessio dynamic and not potential. Kindly refer clearly the main limitation of the sample size, as well as other potential subgroup analysis.
→
Thank you for valuable comment. According to this reviewer’s valuable comment, we reconsider about the study sample size of present study. Even positive tendency between height and active arterial wall thickening among participants without thyroid cysts was observed in present study, the power could not reach significant value. Therefore, to clarify the association between height and active arterial wall thickening among participants without thyroid cysts, larger sample population without thyroid cysts is necessary. However, we already described those limitation in present limitation section.
However, in additional sensitivity analysis, even we found essentially same association between men and women in present results, for women with thyroid cysts, no significant association was observed. Therefore, to perform statistically meaningful sex-specific analysis, further investigation with lager study with women is necessary. Then we also added following sentences in limitation section.
In additional sensitivity analyses, we found essentially the same associations as in the main analysis. In men with thyroid cysts, no significant association between height and active arterial wall thickening was observed. Therefore, to perform statistically meaningful sex-specific analysis, further investigations that involve more men are necessary.
6) Also, a comment of other thyroid pathology and its impact would be of great interest, as the variable of inflammation contributes to the major part of pathological entities.
→
Thank you for valuable comment. According to this reviewer’s valuable comment, we reconsider the potential impact of inflammation which relates to the anti-thyroid peroxidase antibody titer.
Among general Japanese individuals with normal levels of free T3 and free T4, anti-thyroid peroxidase antibody titers were inversely associated with thyroid cysts [10] and active arterial wall thickening [13], which requires CD34-posistive cells [6]. Therefore, low-grade inflammation, which is related to anti-thyroid peroxidase antibody titers, might reduce both the presence of thyroid cysts and endothelial repair activity.

Reviewer 2 Report
As the authors state the absence of data on the presence and concentration of thyroglobulin antibodies is a serious limitation to further their argument in regard to the value of estimating thyroid cysts as a determining factor for hypertension, height and atherosclerosis. They have produced several papers that establish the relationship between TPO Ab and absence of thyroid cysts. In an older age group they have established a relationship between height and the absence of thyroid cysts. They establish in this paper the inverse relationship between arterial wall thickness and thyroid cysts.
The reported thyroid hormone levels are not in SI units and it would be an advantage to the readers to make easy comparisons with other publications. The methodology used IE Roche Abbott etc is not stated nor has the laboratory that produced the results that the paper relies upon been recognized or acknowledged. A common fault with clinical papers.
AT first evaluation there seemed to be an inconsistency in their argument that absence of thyroid cysts indicated a reduced capacity to maintain thyroid function. However there is a logic to their approach in arguing that increased cysts are an indication of the reduced requirement for thyroid activity in old age (indicating increased storage of thyroglobulin, due to reduced demand?) and longevity, hypertension, atherosclerosis and arterial wall thickening. The case for clinical significance of the presence of thyroid cysts has not been unequivocally established but this paper pushes the boundaries further toward the establishment of a relationship that vascular damage (due to hypertension, atherosclerosis and increased intimal thickness) has a role in longevity and height seemed to be a protective factor with increased bone marrow capacity providing a source of CD34+ cells to remodel arterial walls. A determining result of increased CD34+ cells to confirm the hypothesis is also missing and a further study is also required along with thyroglobulin Ab status of the patients. AS the authors also state “the serum concentrations of thyroid hormones are essentially the same among participants with and without thyroid cysts” they rely on an assumption that the presence of cysts reflects a reduced demand for thyroid hormone (increased storage of thyroglubulin?) This does not fit easily with their hypothesis that cysts reflect an increased capacity to meet thyroid hormone demands as the TSH is not significantly increased by the absence of cysts?
Nevertheless, it still is unclear if there is a confounding problem of thyroglobulin antibodies that provides a alternative mechanism. The samples were insufficient to measure the presence of thyroglobulin antibodies. So, a new study would be required to answer that question. Was this planned?
Author Response
Reviewer 2
As the authors state the absence of data on the presence and concentration of thyroglobulin antibodies is a serious limitation to further their argument in regard to the value of estimating thyroid cysts as a determining factor for hypertension, height and atherosclerosis. They have produced several papers that establish the relationship between TPO Ab and absence of thyroid cysts. In an older age group they have established a relationship between height and the absence of thyroid cysts. They establish in this paper the inverse relationship between arterial wall thickness and thyroid cysts.
1) The reported thyroid hormone levels are not in SI units and it would be an advantage to the readers to make easy comparisons with other publications.
→
Thank you for valuable comment. According to this reviewer’s valuable comment, we calculated and changed the hormone units for SI units.
2) The methodology used IE Roche Abbott etc is not stated nor has the laboratory that produced the results that the paper relies upon been recognized or acknowledged. A common fault with clinical papers.
→
Thank you for valuable comment. According to this reviewer’s valuable comment, we re-checked the references and acknowledged of present study. And I confirmed those references and acknowledgement were appropriate.
3-1) AT first evaluation there seemed to be an inconsistency in their argument that absence of thyroid cysts indicated a reduced capacity to maintain thyroid function. However there is a logic to their approach in arguing that increased cysts are an indication of the reduced requirement for thyroid activity in old age (indicating increased storage of thyroglobulin, due to reduced demand?) and longevity, hypertension, atherosclerosis and arterial wall thickening.
→
Thank you for valuable comment. According to this reviewer’s valuable comment, we reconsider the potential influence of thyroid cyst on demand level of thyroid hormone. Since we already descried that ageing is the process of reducing the thyroid demand which might acerates the formation of thyroid cysts, brief description about the association between latent damage of thyroid and thyroid cysts is necessary. Then we added following sentences. And because there is no method to evaluate the demand level of thyroid hormone, development of evaluating the demand levels of thyroid hormone in necessary.
Therefore, among participants without thyroid cysts, even when thyroid hormone production by the entire thyroid gland remains similar to that of participants with thyroid cysts, production per unit of remaining healthy thyroid gland might be higher because of latent thyroid damage. Higher demands for thyroid hormone per unit of remaining healthy thyroid gland might reduce the number and size of thyroid cysts. Further development of methods to evaluate thyroid demand is necessary to clarify the influence of thyroid hormone demand on thyroid cyst formation. Renal function could be a candidate for evaluating the level of demand for thyroid hormone [38].
[38]
Shimizu, Y.; Kawashiri, S.Y.; Noguchi, Y.; Nakamichi, S.; Nagata, Y.; Hayashida, N.; Maeda, T.; Associations among ratio of free triiodothyronine to free thyroxine, chronic kidney disease, and subclinical hypothyroidism. J. Clin. Med. 2022, 11(5), 1269. doi: 10.3390/jcm11051269.
3-2)
The case for clinical significance of the presence of thyroid cysts has not been unequivocally established but this paper pushes the boundaries further toward the establishment of a relationship that vascular damage (due to hypertension, atherosclerosis and increased intimal thickness) has a role in longevity and height seemed to be a protective factor with increased bone marrow capacity providing a source of CD34+ cells to remodel arterial walls.
A determining result of increased CD34+ cells to confirm the hypothesis is also missing and a further study is also required along with thyroglobulin Ab status of the patients. AS the authors also state “the serum concentrations of thyroid hormones are essentially the same among participants with and without thyroid cysts” they rely on an assumption that the presence of cysts reflects a reduced demand for thyroid hormone (increased storage of thyroglubulin?) This does not fit easily with their hypothesis that cysts reflect an increased capacity to meet thyroid hormone demands as the TSH is not significantly increased by the absence of cysts?
→
Thank you for valuable comment. According to this reviewer’s valuable comment, we reconsider the potential mechanism that might underlying the presence of thyroid cysts and CD34-positive cell.
Because the case for clinical significance of the presence of thyroid cysts has not been unequivocally established, present study that showed thyroid cysts could influence on the association between height and active arterial wall thickening is informative as like our studies [ref1][Ref2]. One study reported that status of atherosclerosis could influence on the association between thyroid cysts and hypertension [Ref1]. Another study reported that the status of thyroid cysts could influence on the association between TSH and proteinuria [Ref2]. Then we used essentially same method to make present hypothesis.
[Ref1]
Shimizu, Y.; Kawashiri, S.Y.; Noguchi, Y.; Nagata, Y.; Maeda, T.; Hayashida, N. Association between thyroid cysts and hypertension by atherosclerosis status: a cross-sectional study. Sci Rep 2021, 11(1), 13922. doi: 10.1038/s41598-021-92970-x.
[Ref2]
Shimizu, Y.; Nabeshima-Kimura, Y.; Kawashiri, S.Y.; Noguchi, Y.; Minami, S.; Nagata, Y.; Maeda, T.; Hayashida, N. Association between thyroid-stimulating hormone (TSH) and proteinuria in relation to thyroid cyst in a euthyroid general population. J Physiol Anthropol 2021, 40(1), 15. doi: 10.1186/s40101-021-00264-y.
And the details of the pathology that make us present hypothesis is as following.
Insufficient endothelial repair develops functional atherosclerosis evaluated by CAVI but not structural atherosclerosis evaluated by CIMT [Geriatr Gerontol Int. 2019 Jun;19(6):557-562][ Hypertens Res. 2022 Jun;45(6):1091-1092.]. Active arterial wall thickening is reveled to be positively associated with circulating CD34-positive cell [Sci Rep. 2020 Mar 13;10(1):4656] while inversely associated with anti-thyroid peroxidase antibody titer [ J Clin Med. 2022 20;11(3):521]. Therefore, presence of circulating CD34-postive cell is necessary to progress active arterial wall thickness. And presence of anti-thyroid peroxidase antibody might reduce the circulating CD34-positive cell. In addition to that, anti-thyroid peroxidase antibody titer is inversely associated with thyroid cysts [Environ Health Prev Med. 2020,21;25(1):7]. Participants with absence of thyroid cyst might have disadvantage in endothelial health. Since height is reported to be inversely associated with cardiovascular disease and total death [Int J Epidemiol. 2009;38(4):1060-71] while circulating CD34-positive cell also reported to be inversely associated with cardiovascular disease and total mortality [Circ Res. 2015;116(2):289-297][Atherosclerosis.2021;333:108-115] [Mech Ageing Dev. 2017 Jun;164:139-145.]. Therefore, we thought participants with short stature without thyroid cyst might possess a higher mortality because of the lower productivity of circulating CD34-positive cell count. Therefore, we revised introduction to clarify those above mentioned associations. Our resent hypothesis is evoked from our previous studies.
[Ref9]
Zhao, Y.; Zhang, M; Liu, Y.; Sun, H.; Sun, X.; Yin, Z.; Li, H.; Ren, Y.; Liu, D.; Liu, F.; et al;. Adult height and risk of death from all-cause, cardiovascular, and cancer-specific disease: The Rural Chinese Cohort Study. Nutr. Metab. Cardiovasc. Dis 2019, 29(12), 1299-1307. doi: 10.1016/j.numecd.2019.05.067.
[Ref10]
Patel, R.S.; Li, Q.; Ghasemzadeh, N.; Eapen, D.J.; Moss, L.D.; Janjua, A.U.; Manocha, P.; Kassem, H.A.; Veledar, E.; Samady, H.; et al;. Circulating CD34+ progenitor cells and risk of mortality in a population with coronary artery disease. Circ. Res 2015, 116(2), 289-297. doi: 10.1161/CIRCRESAHA.116.304187.
[Ref11]
Mandraffino, G.; Aragona, C.O.; Basile, G.; Cairo, V.; Mamone, F.; Morace, C.; D'Ascola, A.; Alibrandi, A.; Lo, Gullo, A.; Loddo, S.; et al;. CD34+ cell count predicts long lasting life in the oldest old. Mech. Ageing. Dev 2017, 164, 139-145. doi: 10.1016/j.mad.2017.03.003.
Then we added following sentences in introduction section.
In addition, thyroid hormone regulates the proliferation of CD34-positive cells [9]. The existence of thyroid cysts could indicate the absence of latent thyroid damage [10–12]. Anti-thyroid peroxidase antibodies are known to decrease thyroid hormone synthesis. A cross-sectional study of Japanese with normal thyroid function reported an inverse association between anti-thyroid peroxidase antibody titer and thyroid cysts [10]. Anti-thyroid peroxidase antibody titers were also reported to be inversely associated with active arterial wall thickening [13]. Therefore, participants without thyroid cysts might have worse endothelial health. In other words, among older individuals, the presence of thyroid cysts could indicate a relatively higher capacity for producing CD34-positive cells.
Since height has been reported to be inversely associated with death from all causes and cardiovascular disease [14] and circulating CD34-positive cells have been reported to be inversely associated with cardiovascular disease and all-cause mortality [15,16], understanding the influence of thyroid cysts on the association between height and active arterial wall thickening could help clarify a novel mechanism in vascular remodeling.
3-3)
Nevertheless, it still is unclear if there is a confounding problem of thyroglobulin antibodies that provides a alternative mechanism. The samples were insufficient to measure the presence of thyroglobulin antibodies. So, a new study would be required to answer that question. Was this planned?
→
Thank you for valuable comment. According to this reviewer’s valuable comment, we reconsider the meaning of evaluating the thyroglobulin antibodies on present study.
Unfortunately, because of the shortage of serum sample, we could not measure anti-thyroglobulin. However, present study is aimed to clarify the thyroid cysts specific association between height and active arterial wall thickening. And we thought the latent damage of thyroid might underling present results. Therefore, thyroid cyst itself but not autoimmune antibody is the important determinant on present associations.
To evaluate the association between autoimmune antibody and thyroid cysts, the information of thyroglobulin antibodies became important because antibody could influence on forming thyroid cysts. And we previously reported inverse association between thyroid peroxidase anti-body and thyroid cysts even the influence of anti-thyroglobulin antibody could not be adjusted. However, since inverse association between thyroid peroxidase antibody and thyroid cysts were observed, this inverse association should be weekend by the presence of anti-thyroid globulin. Therefore, because unadjusted for antithyroid globulin model weaken the association between thyroid cyst and anti-peroxidase antibody and thyroid cysts while significant inverse association between anti-peroxidase antibody and thyroid cyst were observed in previous study, data of anti-thyroid antibody is not so important for present study as like our previous studies [Ref1][Ref2].
[Ref1]
Shimizu, Y.; Kawashiri, S.Y.; Noguchi, Y.; Nagata, Y.; Maeda, T.; Hayashida, N. Association between thyroid cysts and hypertension by atherosclerosis status: a cross-sectional study. Sci Rep 2021, 11(1), 13922. doi: 10.1038/s41598-021-92970-x.
[Ref2]
Shimizu, Y.; Nabeshima-Kimura, Y.; Kawashiri, S.Y.; Noguchi, Y.; Minami, S.; Nagata, Y.; Maeda, T.; Hayashida, N. Association between thyroid-stimulating hormone (TSH) and proteinuria in relation to thyroid cyst in a euthyroid general population. J Physiol Anthropol 2021, 40(1), 15. doi: 10.1186/s40101-021-00264-y.
